# MIL-53 Metal–Organic Framework as a Flexible Cathode for Lithium-Oxygen Batteries

**DOI:** 10.3390/ma14164618

**Published:** 2021-08-17

**Authors:** Yujie Zhang, Ben Gikonyo, Hicham Khodja, Magali Gauthier, Eddy Foy, Bernard Goetz, Christian Serre, Servane Coste Leconte, Vanessa Pimenta, Suzy Surblé

**Affiliations:** 1Université Paris-Saclay, CEA, CNRS, NIMBE, 91191 Gif-sur-Yvette, France; yujie.zhang@cea.fr (Y.Z.); ben.gikonyo@univ-lyon1.fr (B.G.); hicham.khodja@cea.fr (H.K.); magali.gauthier@cea.fr (M.G.); eddy.foy@cea.fr (E.F.); 2Laboratoire des Multimatériaux et Interfaces, Université Claude Bernard Lyon 1, UMR CNRS 5615, 69622 Villeurbanne, France; 3Institut des Matériaux Poreux de Paris (IMAP), ESPCI Paris, Ecole Normale Supérieure de Paris, CNRS, PSL University, 75005 Paris, France; bernard.goetz@espci.fr (B.G.); christian.serre@espci.fr (C.S.); vanessa.pereira-pimenta@espci.fr (V.P.); 4INSTN, Ecole de spécialisation des énergies bas carbone et des technologies de la santé, Unité d’Enseignement de Saclay, CEA, 91191 Gif-sur-Yvette, France; servane.coste-leconte@cea.fr

**Keywords:** metal–organic frameworks, lithium-oxygen batteries, electrochemical performance

## Abstract

Li-air batteries possess higher specific energies than the current Li-ion batteries. Major drawbacks of the air cathode include the sluggish kinetics of the oxygen reduction (OER), high overpotentials and pore clogging during discharge processes. Metal–Organic Frameworks (MOFs) appear as promising materials because of their high surface areas, tailorable pore sizes and catalytic centers. In this work, we propose to use, for the first time, aluminum terephthalate (well known as MIL-53) as a flexible air cathode for Li-O_2_ batteries. This compound was synthetized through hydrothermal and microwave-assisted routes, leading to different particle sizes with different aspect ratios. The electrochemical properties of both materials seem to be equivalent. Several behaviors are observed depending on the initial value of the first discharge capacity. When the first discharge capacity is higher, no OER occurs, leading to a fast decrease in the capacity during cycling. The nature and the morphology of the discharge products are investigated using ex situ analysis (XRD, SEM and XPS). For both MIL-53 materials, lithium peroxide Li_2_O_2_ is found as the main discharge product. A morphological evolution of the Li_2_O_2_ particles occurs upon cycling (stacked thin plates, toroids or pseudo-spheres).

## 1. Introduction

While emissions of greenhouse gases are reaching alarming levels, the development of alternative energies is even more important, particularly in the automotive field. Up to now, the lithium-ion battery holds a prominent place in the battery field. Nevertheless, although its energy densities are still increasing and its cycle life exceeds thousands of cycles, its specific capacity and energy density seem to reach their limits and will be insufficient for the long-term needs. Lithium-air batteries are the object of growing interest nowadays, due to the much higher theoretical gravimetric energy storage density of Li-O_2_ systems compared to other technologies [1,2]. The non-aqueous lithium-air cell consists of a lithium metal anode, an electrolyte for Li^+^ conduction and a porous air cathode. The porous electrode is exposed to ambient air to store and convert energy. There are numerous scientific and technical challenges that must be overcome for Li-air batteries: capacity fading during cycling, electrolyte instability or large discharge and charge overpotentials [3]. In this paper, we focus primarily on the drawbacks of air cathodes [4,5].

During the discharging process, the reaction between Li^+^ and O_2_ at the cathode leads mainly to the formation of lithium peroxide (Li_2_O_2_) [6], although lithium oxide (Li_2_O) can also be found. In non-aqueous Li-O_2_ batteries, the reaction is depicted as 2 Li^+^ + 2 e^−^ + O_2_ → Li_2_O_2_ (E° = 2.96 V vs. Li^+^/Li), well known as the Oxygen Reduction Reaction (ORR). During the charge process, the reversible process theoretically takes place, leading to the decomposition of Li_2_O_2_ and the release of O_2_ (Oxygen Evolution Reaction, OER). The kinetics of ORR are sluggish, leading to high overpotential and poor cyclability. The insoluble Li_2_O_2_ discharge products precipitate and are stored in the pores of the air cathode, and may further block the oxygen transport. Furthermore, the end of the discharge generally occurs when the porosity of the cathode (usually porous carbon) is clogged or when the discharge products block the access to the pores. This leads to a capacity drop, with values lower than the theoretical capacity [7]. The decomposition of the porous carbon is also observed in the presence of the insulating Li_2_O_2_ [8]. Hence, it appears essential to develop new cathode materials, which are able to accommodate a substantial amount of Li_2_O_2_ without blocking the cathode pores. The pore size and volume appear as key parameters that may directly affect the discharge capacity of the material [9].

Controlling the particle size, the morphology and the uniformity of the porous space is, thus, of great importance in the development of new air cathodes. Many studies have focused on the development of porous carbon electrodes owing to their carbon high conductivity, high surface area and low cost [10]. Above all, porous air electrodes must be able to efficiently catalyze both the oxygen reduction reaction (ORR—taking place at the air electrode upon discharge) and its reverse reaction (oxygen evolution reaction (OER)—during the charging process). The addition of a metal or oxide catalyst into the air-cathode is also a way to improve the efficiency of the Li-air batteries [11,12,13].

Metal–Organic Frameworks (MOFs) have recently attracted much attention for energy related applications, but also due to their catalyst properties [14,15]. MOFs are a class of porous materials consisting of metal ions coordinated with organic ligands through covalent bonds. These hybrid solids are highly ordered crystalline materials with a very large structural (pore sizes/shapes) and chemical diversity (functional acid or basic groups, tunable hydrophilic/hydrophobic balance) [16]. Electrochemical applications of MOFs, especially as electrode materials in Li-ion and Li-S batteries, have been developed recently and have exhibited tremendous potential due to their abundant electroactive sites and large ion diffusion channels [17]. More recently, pristine MOFs and MOF-derivatives appeared as promising candidates as air electrodes for Li-O_2_ batteries [18]. During the charge/discharge processes, their open structure could provide a host network for both lithium ion and oxygen diffusion and a good diffusion of oxygen, while the high surface area could accommodate the discharge products. Sacrificial MOFs or MOF-derivatives materials are also good catalysts. They are obtained from pyrolysis of pristine MOF, resulting in metal or metal oxide nanoparticles immobilized in carbonaceous nanostructures. Thanks to the initial MOF structure, the catalytic sites in these powders are then more homogenously distributed. Up to now, few pristine MOFs have been studied as air cathode materials for Li-air/Li-O_2_. Li et al. have, for the first time, reported the electrochemical properties of several benchmark MOFs with high surface area (HKUST-1, MOF-5 and M-MOF-74 with M=Mg, Co, Mn) [19]. A first discharge capacity of 9420 mAh/g was achieved with Mn-MOF-74, while lower capacities were observed for the isostructural M-MOF-74 (4560 mAh/g and 3630 mAh/g, respectively, for Mg and Co). It should, however, be noted that these capacities were obtained using a high content of super P carbon additive (40 wt.%) to increase the conductivity.

We propose, for the first time, to study the electrochemical properties of a flexible MOF material as an air cathode for Li-O_2_ batteries. It is expected that the flexibility of the structure will make it possible to easily accommodate the discharge products into the cathode pores, avoiding the clogging of pores. It is well-known that the dynamic of adsorption/desorption of guest molecules in flexible MOF structures is different compared to a rigid framework [20,21]. Aluminum terephthalate—MIL-53(Al) [22]—has recently been highlighted as a potential air-cathode material for Al-air systems; it exhibited highly stable cyclic voltammetry behavior for Al-air applications [23]. This material is particularly interesting for this study, not only due to its flexibility but also due to the high chemical stability of the framework and inert redox activity of the metal used to construct the framework (Al), avoiding all parallel redox reactions. In addition, the material is low cost, and easy to synthesize and scale-up. Its structure is known for its high flexibility, while keeping its mechanical properties. Herein, we compare the electrochemical performance of MIL-53(Al) synthetized by hydrothermal or microwave routes. Ex situ XRD, SEM and XPS were performed in order to highlight the nature and the morphology of the discharge products.

## 2. Materials and Methods

### 2.1. Preparation of MIL-53(Al)

Al(OH)[O_2_C-C_6_H_4_-CO_2_] or MIL-53 (MIL stands for Materials Institute of Lavoisier) were prepared using environmentally friendly syntheses. All chemical reactants, including aluminum nitrate nonahydrate (Al(NO_3_)_3_.9H_2_O, 99+%, Alfa Aesar (Thermo Fisher GmbH, Kandel, Germany) and benzene 1,4-dicarboxylic acid (C_6_H_4_-1,4-(COOH)_2_, 99+%, Alfa Aesar, abbreviated as BDC hereafter), were used as purchased. Deionized water was prepared by a Milli-Q PF ultrapure water system.

MIL-53(Al) was first synthesized under hydrothermal conditions using the previous procedure described by Loiseau [22] at a scale-up of 5.4 times (called hereafter H-MIL-53). The molar composition of the starting gel used was 2:1:160 for Al salt, BDC and deionized water, respectively. The reaction was performed in a 125 mL Teflon-lined stainless-steel Parr autoclave, under autogenous pressure for three days at 220 °C and cooled down to room temperature. The light white solid was filtered, washed and dried at room temperature, leading to the as-synthesized solid.

Synthesis using microwave-assisted heating was carried out using molar ratios equivalent to the conventional hydrothermal conditions (hereafter referred to as MW-MIL-53). Typically, 13.1 mmol of Al(NO_3_)_3_.9H_2_O, 6.6 mmol of BDC and 19 mL of deionized water were mixed together in a 95 mL Teflon-liner and stirred for 45 min. The resulting mixture solution was then transferred into a Teflon microwave reactor and heated at 220 °C for 30 min (temperature’s rate 2 °C/s) in a Mars 6 microwave, CEM. After the reaction, the resulting white suspension was centrifuged and the obtained product was washed and dried at room temperature to obtain the as-synthesized solid.

Both the H-MIL-53 and MW-MIL-53 solids were finally calcined at 360 °C for 13 h to eliminate the non-reacted linker molecules which remain in the pores after synthesis.

### 2.2. Characterizations of Pristine MOFs

X-ray diffraction patterns of MIL53 were collected on a powder sample with Bruker D8 Advance (Bruker, Karlsruhe, Germany) and Siemens D5000 (θ–2θ mode) diffractometers (Bruker, Karlsruhe, Germany) using Cu-Kα radiation (λ 1.5406 Å and λ = 1.5444 Å) in the 2θ range of 5–40° (available, respectively, at ESPCI and CEA-INSTN). The flexibility of MIL-53 was investigated upon solvent adsorption/desorption or guest encapsulation. XRD patterns were indexed using Dicvol program [24] and pattern matching refinements were performed with Fullprof Software (version April 2019, The Fullprof suite, Rennes, France) [25]. Scanning electron microscopy (SEM) images were recorded using a field emission scanning electron micro-analyzer (FEI Magellan 400) at ESPCI (Hillsboro, OR, USA). Thermogravimetric analysis (TGA) was performed under air atmosphere (3 °C/min) using a Mettler Toledo TGA/DSC 2, STAR System from room temperature to 800 °C. The surface area of the dehydrated samples was measured by nitrogen porosimetry using a Micromeritics Tristar (Martignas, France) instrument at 77 K, after being activated at 200 °C for 12 h in a Smart VacPrep™ 067 apparatus.

### 2.3. Battery Assembly and Testing

MIL-53 powders were first calcined at 200 °C for one day in order to obtain large pore phases in the starting materials (MIL-53-lp). Composite electrodes were prepared by mixing the MIL-53-lp active material, Super P carbon black (TIMCAL, 99+%, Alfa Aesar) and poly(vinylidene fluoride) (PVDF, Kynar 2801) at a weight ratio 65/25/10 with N-methyl-2-pyrrolidone (NMP, 99.8%, Acros Organics, Thermo Fisher GmbH, Strasbourg, France). The resulting slurry was coated onto a Toray carbon paper (TGP-H-60, Alfa Aesar, Thermo Fisher GmbH, Kandel, Germany) as the O_2_ electrode Electrodes were first dried at room temperature for 2 h and then at 80 °C under vacuum for 24 h. As a comparison, carbon black electrodes were prepared through a similar procedure (90 wt.% of Super P carbon, 10 wt.% PVDF). Li-O_2_ cells were assembled in an argon-filled dry glovebox using an ECC-Air electrochemical cell (El-Cell GmbH, Hambourg, Germany) configuration with openings allowing oxygen to enter. The battery cell consists of a disc of composite electrode (φ = 18 mm) as an air cathode, 2 borosilicate glass-fiber separators (φ = 18 mm, thickness 0.26 mm, GF-A Whatman (Dutscher, Bernolsheim, France) and an Li foil (φ = 18 mm, thickness 0.38 mm, 99.9%, Sigma-Aldrich, Saint Quentin Fallavier, France) (Figure 1). The electrode mass loading (MIL-53 and Super P carbon, called active material AM) was around 1 mg/cm^2^; hereafter the unit capacity is in mAh/g_AM_. The non-aqueous electrolyte was 1M LiTFSI (lithium bis[trifluoromethanesulfonyl imide], 99.95%, Sigma-Aldrich) in 1,2-dimethoxyethane (DME, >99.5%, Acros Organics). After assembly, the cells were purged with pure O_2_ at 0.3 bar for 5 min. To obtain a slightly positive pressure inside the cell, the inlet was kept opened for 10 s longer than the gas outlet.

Electrochemical characterizations were carried out on both pure carbon black and MOF air cathodes using a Biologic VMP-300 potentiostat (Biologic Science Instruments, Grenoble, France) at room temperature. Galvanostatic cycling was performed at a constant current density of 50 mA/g_AM_ in the voltage range of 2.0–4.5 V vs. Li^+^/Li. The cells were aged for 6 h of rest at open circuit potential (OCV) before measurements. The capacity values are reported hereafter with respect to the combined mass of MIL-53 and Super P carbon.

### 2.4. Characterizations after Cycling

Immediately after the cycling, Li-O_2_ cells (Super P carbon and H-MIL-53) were disassembled in an argon-filled glovebox and the discharged electrodes collected for ex situ characterizations. The discharged electrodes were gently washed with a few drops of DME to eliminate electrolyte residues. Each electrode was sectioned with a scalpel into four smaller samples for analysis.

To avoid any reactivity with air, the samples were sealed in Kapton tape in the glovebox prior to XRD analysis with a RU-200B rotating anode X-ray generator (Rigaku, Neu-Isenburg, Germany) using Kα-Mo radiation (λ = 0.7093 Å) in transmission mode at CEA-NIMBE. Diffracted patterns were recorded using a RebirX-70S (Cegitek, Aubagne, France) hybrid pixel array detector.

The morphology of the solid products formed during the discharge reaction were studied by Scanning Electron Microscopy using a SEM-FEG Carl Zeiss Ultra 55 at CEA-NIMBE (Oberkochen, Germany). The samples were prepared in the glovebox and then transferred quickly to the microscope chamber. Air exposure occurred for a period of a few seconds, which may not have changed the morphology of the sample.

Pristine and discharged electrodes were characterized with X-ray photoelectron spectroscopy (XPS) using a Kratos Analytical Axis Ultra DLD (Kratos Analytical, Manchester, UK) with monochromatic Al Kα excitation (1486.7 eV) and a charge neutralizer. All spectra were recorded with a pass energy of 40 eV. The cycled electrodes were loaded into the XPS apparatus without exposition to ambient air using a sample transfer vessel. All spectra were calibrated with the C 1s photoemission peak of adventitious carbon at 284.8 eV. After subtraction of a Shirley-type background, photoemission lines were fitted using combined Gaussian–Lorentzian functions.

## 3. Results and Discussion

### 3.1. Pristine MOF Materials Characterizations

The synthesis of MIL-53(Al) was performed under hydrothermal conditions, using conventional or microwave assisted methods (H-MIL-53 and MW-MIL-53, respectively). The synthesis routes were varied to study their effect on the particles’ size, morphology and crystallinity, and how these parameters are related to their electrochemical properties.

MIL-53(Al) consists of AlO_4_(OH)_2_ corner-sharing octahedral chains interconnected by 1,4-benzenedicarboxylate groups to form a three-dimensional network with one dimension rhombic-shaped pores. This framework is well known for its breathing behavior, with structural transitions from a large pore (MIL-53-lp) or high temperature phase to a narrow pore (MIL-53-np) or low temperature phase [22]. Figure 2 shows the pore configuration of the three MIL-53 crystal structures and their breathing effect. The pores of the as-synthesized form (MIL-53-as) are filled with disordered free acid. The large pore form, which corresponds to a fully empty pore configuration, can only be isolated under controlled atmosphere and high temperature. As the framework easily adsorbs water molecules from the atmosphere, the narrow pores phase (or np) is the most commonly isolated form. Moreover, a mixture of both forms is often observed in PXRD patterns.

The X-ray diffraction patterns of the as-synthetized samples (Appendix A) correspond perfectly to the as-synthetized form described earlier by Loiseau and co-workers [22]. TGA measurements confirm that the remaining 1,4-benzendicarboxylic (BDC) acid adsorbed in the pores is effectively removed by the overnight heat treatment at 360 °C (Appendix A). The particle size and morphology of H-MIL-53 and MW-MIL-53 were investigated by SEM for the MIL-53-np form of both samples (Figure 3). With the hydrothermal synthesis, micrometer-sized platelets (average crystal length of ~2 μm) are obtained, while small pseudospherical particles (<1 μm) are observed for the microwave synthesis. We see clearly that the length-width or aspect ratio of the particles is smaller for these later synthesis routes. The microwave irradiation allows a very fast synthesis of MIL-53 materials (here 30 min. instead of 3 days at 220 °C for H-MIL-53). It generates smaller particles than the conventional hydrothermal synthesis (ca. 4 times lower) as the nucleation process is favored instead of the growth of crystals as in conventional hydrothermal method.

The effect of size in flexible MOFs is known to strongly impact the physical and structural properties of the material [26]. Downsizing the particles size directly influences the breathing effect of the framework, usually making it easier once the particle size decreases, enhancing the diffusion of the trapped species in the pores. In addition, flexibility is also influenced by the interaction of the framework with solvents [27]. One can, therefore, expect a better diffusion of both the electrolyte and the discharge products when considering smaller particles. One other aspect to be considered is the outer surface chemistry, which is different in nano-sized and bulk materials [28]. The interaction between the MOF particles and the other components of the electrodes (carbon and polymers) is expected to be different, with a stronger influence when nanoparticles are in play, and therefore, to have an influence on the electrochemical performance.

The surface areas of both MIL-53 and Super P carbon, determined by the Brunauer–Emmett–Teller (BET) method, and the external surfaces obtained through the Harkins–Jura equation, are summarized in Table 1. These values are in agreement with those already reported [22]. As expected, MIL-53 samples show high porosity compared to the Super P carbon. Both MIL-53 compounds display a high surface area, with MW-MIL-53 having a slightly higher value, which is certainly related to the smaller particle size. Although the external surface area of both compounds is in the same range, we can note that the ratio between the external surface area and the overall surface area decreases from Super P to MW-MIL-53. This will most likely influence the electrochemical performance, since the external surface area actively participates in the redox process.

### 3.2. Electrochemical Properties

The XRD pattern of the pristine electrode shows that the breathing transition occurred during the electrode preparation (Figure 4) and corresponds perfectly to the PXRD diagram recorded on MIL-53 powder impregnated with the binder (10 wt.% PVDF in DME) then dried at 80 °C under vacuum.

The structural transformations of MIL-53 powders, when exposed to the solvents used during the electrode preparation or during the cycling, were investigated (Appendix A). The evolution of the cell parameters obtained by pattern matching are given in Appendix A. No change was observed when the electrode was exposed to the electrolyte (Appendix A).

Numerous independent cells with H-MIL-53/MW-MIL-53 were cycled for reproducibility and to establish the average capacity of each system. Figure 5 reports the first and second discharge capacities of all the Li-O_2_ batteries tested with H-MIL-53 or MW-MIL-53. Several trends are observed for the first discharge: the capacities are either around 800–1200 mAh/g_AM_ (60% of electrochemical tests) or present very limited values near 40–100 mAh/g_AM_ (40% of electrochemical tests). In the latter case, the second discharge capacities are higher than the first ones, denoting a possible activation process during the first cycle. We can infer that in some electrodes, an initial activation cycle is necessary to remove a possible passivation layer or to remove some solvent molecules, present in the pores, that may at first prevent the nucleation of the discharge products. The open circuit voltages (OCV) are lower when the second discharge capacities are higher (2.83–3.06 V and 2.77–2.81 V). Examples of galvanostatic cycling are represented on Appendix A for both MIL-53 electrodes. Despite the difference in terms of particle size for the two compounds H-MIL-53 and MW-MIL-53, their electrochemical properties seem to be equivalent. MIL-53 electrodes display a lower capacity when compared to Super P electrodes (3143 mAh/g_AM_, see Appendix A), certainly due to the insulating character of the MOF, which directly decreases the electrode conductivity.

Figure 6 shows the performance of the discharge and charge capacities upon cycling. Both H-MIL-53 and MW-MIL-53 compounds show an equivalent electrochemical behavior, suggesting in the first instance that the size of the particles does not play a key role in the electrochemical behavior, as was expected. In the case where the first discharge capacity is low and the second discharge is high (case 1), we observe the highest charge capacities at cycle 2 (800–1000 mAh/g_AM_); however, there is a severe drop in capacity after the fourth cycle. On the other hand, when the discharge capacity quickly decreases over cycling (case 2), no charge capacity is observed, the OER reaction does not occur for both MIL-53 compounds. Finally, an additional and third trend can be observed only for the H-MIL-53 compound (case 3): if the first discharge capacity is medium (~800 mAh/g_AM_), we observe a slow decrease in discharge and charge capacities.

Figure 7 compares the discharge and charge potentials at the first and second cycles. The discharge potential is slightly higher for Super P carbon electrode (~2.72 V) than for the MIL-53 electrodes (centered on 2.60 V) for all cases mentioned above. This may denote a possible better conductivity of the carbon electrode compared to the more insulating MOFs-based electrodes. For all samples, a single plateau is observed with a continuous slight slope (2.55–2.67 V) at the first cycle. All discharge potential values are in agreement with the literature [29] and suggest the formation of Li_2_O_2_ as the main discharge product. For the second discharge, the profiles are different: a single plateau similar to the first discharge profile (~2.60 V, case 1 and case 3), or two successive plateaus (occurring at around 2.58–2.61 V and 2.48–2.51 V, case 2), are observed. The lower potential for MOF materials may suggest some limitations in the transfer of species inside the electrodes or an obstruction of catalytic sites by the discharge products. While the overpotential is larger for MOF electrodes, the polarization is still limited with regard to the thermodynamic potential for the formation of Li_2_O_2_ (E_Li2O2_ = 2.96 V) [30]. The poor electric conductivity of MOF also increases the polarization and causes a high overpotential, leading to unsatisfactory ORR and OER reactions [31].

The charge overpotential is also higher for both MIL-53 than for Super P carbon electrodes (Figure 7b). When the second discharge capacity is higher than the first one (case 1, dash line), we observe a single plateau (centered on 4.46 V) for the second cycle and for the following cycles. When the first discharge capacity is higher (case 2, blue or red solid line), the cells very quickly reach the potential limit (4.5 V) fixed by the stability window of LiTFSI in the DME electrolyte, thus preventing the complete OER reaction. This high overpotential most probably explains the decrease in capacity over time. The charge profile for the case 3 (medium first discharge capacity, black line) is slightly different from the other cases; we observe a single plateau with a continuous slight slope (4.45–4.48 V). One might believe that this behavior is related to the accessible external area, which enhances interface phenomena. The high internal surface area actively participates in the storage of the discharge products, therefore leading to a better electrochemical performance.

### 3.3. Ex Situ Characterizations

In order to obtain more insights on the electrochemical mechanisms in the MOF-based electrodes, the nature and the morphology of the discharge products for the H-MIL-53 electrodes were investigated by XRD, XPS and SEM.

Figure 8 shows the ex situ XRD patterns collected after the first and the tenth discharges for H-MIL-53 electrodes with a higher initial capacity. By comparing the XRD patterns of the Super P and H-MIL-53 electrodes, we can attribute the first 2θ position to H-MIL-53. We observe a shift of the 2θ value upon discharge (from 4.17° to 3.85°, respectively, for pristine and discharged electrodes), which evidences a pore configuration evolution of H-MIL-53 while accommodating the discharge products inside the pores. More importantly, as suggested by the electrochemical data, lithium peroxideLi_2_O_2_ is found as the main discharge product (Bragg peaks (101) and (110)) for both MIL-53 electrodes (Appendix A). The intensity of the main peak at 15.7° increases with the number of charge–discharge cycles for both electrodes, denoting the accumulation of Li_2_O_2_ products upon cycling.

To reveal the morphology of the discharge products, the cycled electrodes with both MIL-53 compounds were also observed by SEM after the first and tenth discharges (Figure 9). For the first discharge, we observe small toroidal crystals (surrounded in yellow) or thin plates (surrounded in pink) for H-MIL-53 and MW-MIL-53, respectively. After 10 discharges, the toroids strongly evolve and present a shape close to Li_2_O_2_ spheres for both MIL-53 electrodes, while small toroids are still observed. The first discharge capacity of MW-MIL-53 is lower that the first capacities for ex situ cycled electrodes (Appendix A), that is in agreement with the presence of large toroidal particles [32]. One might suppose that the Li_2_O_2_ thin plates lead to the development of toroid particles and then sphere particles [33]. Assuredly, both SEM and XRD results demonstrate that Li_2_O_2_ is the main discharge product formed when the H-MIL-53 electrodes are used as cathodes.

The electrode surface’s composition in the discharged H-MIL-53 electrodes was further investigated by X-ray photoelectron spectroscopy. Figure 10 compares the C 1s, S 2p, F 1s, O 1s and Li 1s core peaks collected on the pristine and discharged electrodes (1 and 10 discharges). In the C 1s region, for the pristine electrode, the contribution at approximatively 285 eV is fitted into two contributions corresponding to sp^2^ C=C (284.4 eV) and sp^3^ C-C (284.8 eV) in the benzene ring of the terephthalate ligand. The latter peak may also come from the adventitious carbon. Peaks at 286.0 eV and 288.7 eV binding energies are indexed to the C-O and O-C=O bonds, respectively. These peaks are in accordance with the organic linker in the MIL-53 reported in the literature [34]. Additional peaks are observed for the pristine electrode and are attributed to CF_2_ bonds (291.3 eV) from the PVDF binder [35]. After 1 or 10 discharges, lithium peroxide (Li_2_O_2_) is clearly confirmed as the main discharge product, as shown by the peaks at 54.9 eV and 531.5 eV in the Li 1s and O 1s regions, respectively. While the discharged electrodes were rinsed with some drops of DME solvent to remove the excess electrolyte, we still see some contributions from the LiTFSI salt (CF_3_ at 293 eV, 689 eV in the F 1s region, and the first doublet peaks at 169.3 eV and 170.1 eV in the S 2p region assigned to S 2p_3/2_ and S 2p_1/2_ of the LiTFSI salt [36]). Interestingly, an additional doublet (167.1 eV and 168.3 eV) is observed in the S 2p region for the first discharge electrode, corresponding to S=O bonds, most probably arising from decomposition products of the electrolyte salt [37].

## 4. Conclusions

For the first time, we investigated the electrochemical properties of a flexible Metal–Organic Framework in Li-O_2_ systems. MIL-53 compounds were synthetized through hydrothermal and microwave-assisted routes (H-MIL-53 and MW-MIL-53, respectively) in order to investigate the potential impact of the particle size/aspect ratio on their electrochemical behavior. Microwave irradiation leads to a smaller particle size than hydrothermal synthesis (ca. 4 times lower). The SEM images reveal different morphologies with micrometer-sized platelets for H-MIL-53 and small pseudospherical particles for MW-MIL-53. Both MIL-53 samples have higher BET surfaces than the Super P carbon black, while the external surfaces are similar.

The MIL-53 materials’ electrochemical behavior was investigated in composite electrodes containing carbon and compared with pure carbon electrodes. We showed herein that the MIL-53(Al) material is an efficient air cathode material. Ex situ experiments (XRD, SEM and XPS) performed after discharging evidenced that lithium peroxide Li_2_O_2_ is the main discharge product stored in the H-MIL-53 material. The morphology of the Li_2_O_2_ evolved upon cycling: small toroidal crystals or thin plates are present at the first discharge, whereas Li_2_O_2_ spheres and toroidal particles are observed after 10 discharges. The MOF materials show interesting discharge capacities; however, the expected impact of the size/aspect ratio on the electrochemical properties could not be demonstrated, as both the H-MIL-53 and MW-MIL-53 electrodes showed similar capacities for the first discharge (around 1000 mAh/g_AM_). Moreover, higher overpotentials for charge and discharge processes are observed for both MIL-53 electrodes compared to the Super P carbon ones, suggesting some limitations in the electrical conductivity of the MOF materials or some limitations in the transport of species inside the electrodes. The low performance of MIL-53 can be explained by two main factors: (1) the lack of reactive open metal sites within the MIL-53 structure and (2) the low electronic conductivity of the MIL-53 electrodes (40 wt.% carbon used in the previous examples instead of the 25 wt.% used in our work). Accordingly, to overcome the MOFs’ conductivity drawback, we envision the design of new materials, in particular the synthesis of MOF/C composites including highly conductive agents such as Ketjenblack or graphene oxide [38,39]. Combining the high conductivity of carbon with the high porosity and flexibility of the MOF could be an efficient way to enhance the electrochemical properties of MIL-53 electrodes for Li-O_2_ batteries.

## Figures and Tables

**Figure 1 materials-14-04618-f001:**
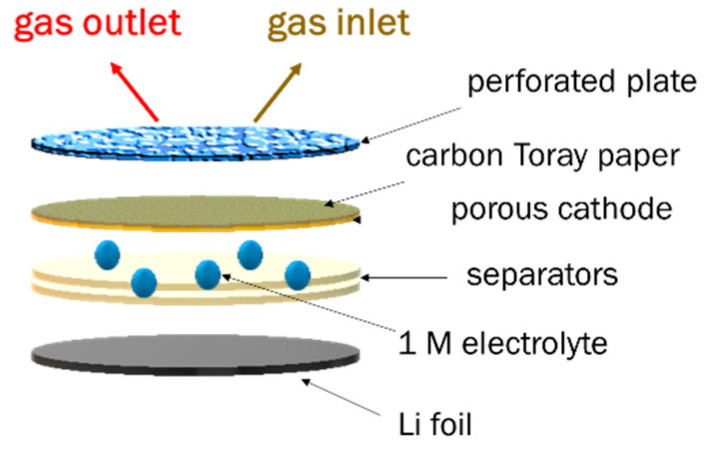
Schematic representation of the Li-air battery.

**Figure 2 materials-14-04618-f002:**
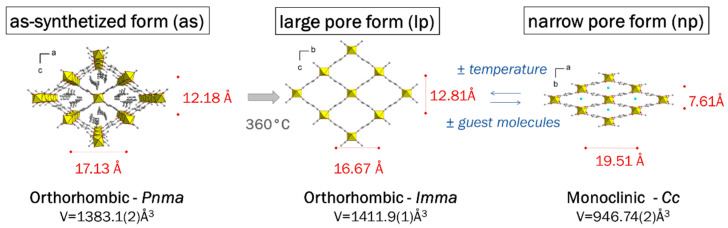
Structural transformation of the MIL-53 framework between its as-synthesized (as) (with free organic linker in the pore), open (lp) and contracted (np) forms (with water molecules in the pore). Aluminum octahedra, oxygen, carbon atoms and guest molecules are represented in yellow, red, grey and cyan, respectively.

**Figure 3 materials-14-04618-f003:**
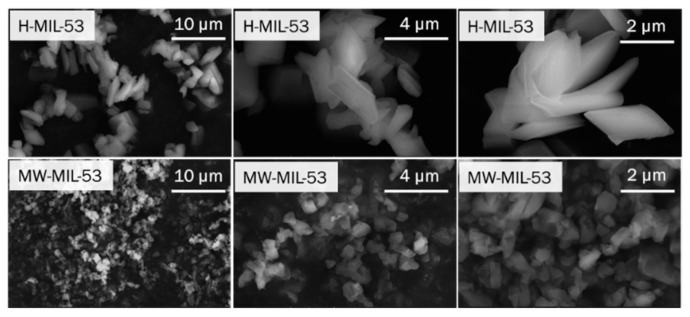
(From left to right) low, medium and high magnification SEM images of (top) H-MIL-53 and (bottom) MW-MIL-53.

**Figure 4 materials-14-04618-f004:**
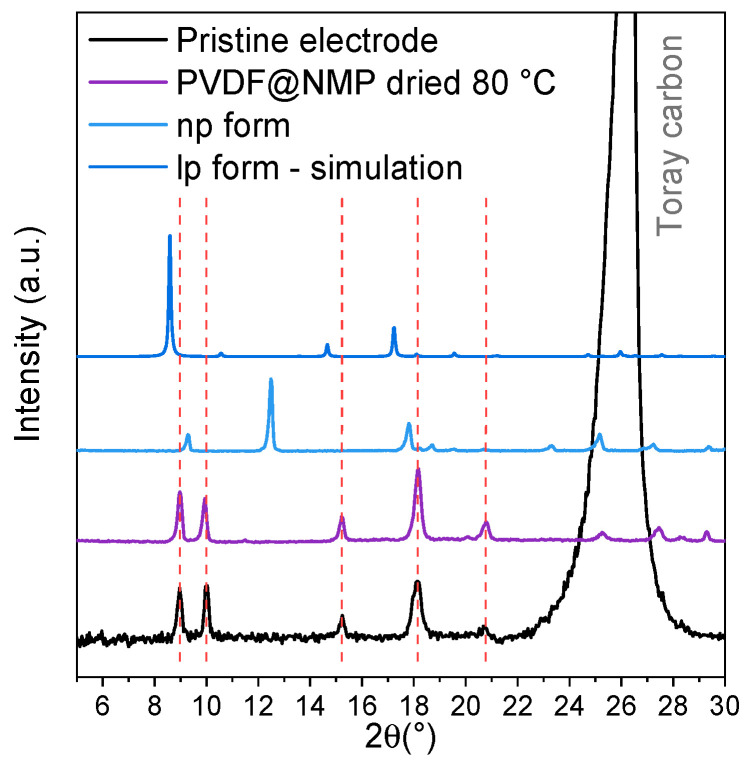
Comparison of XRD patterns (Cu-Kα) of pristine electrode (black) and various forms of MIL-53 powders (violet: MIL-53 with PVDF@NMP dried at 80 °C; blue: np form; dark blue: lp form).

**Figure 5 materials-14-04618-f005:**
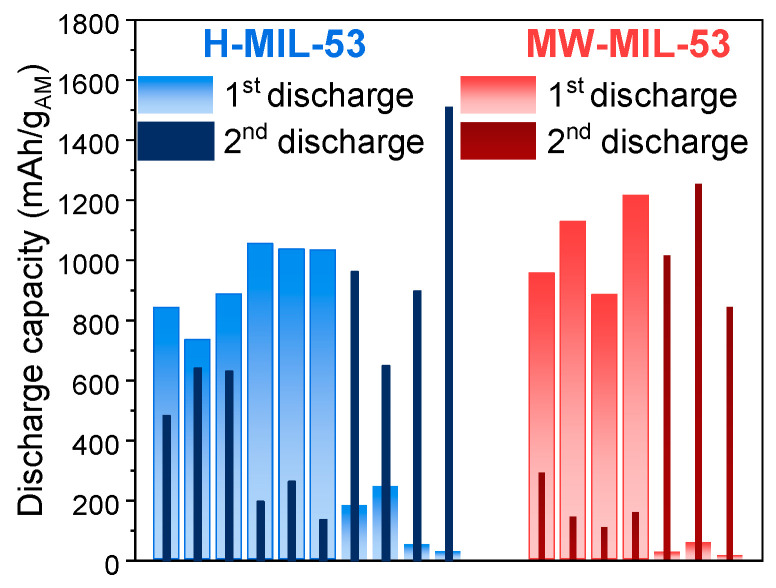
Discharge capacities at the first (light color bars) and second (dark color bars) discharges for all Li-O_2_ batteries with H-MIL-53 (blue) or MW-MIL-53 (red).

**Figure 6 materials-14-04618-f006:**
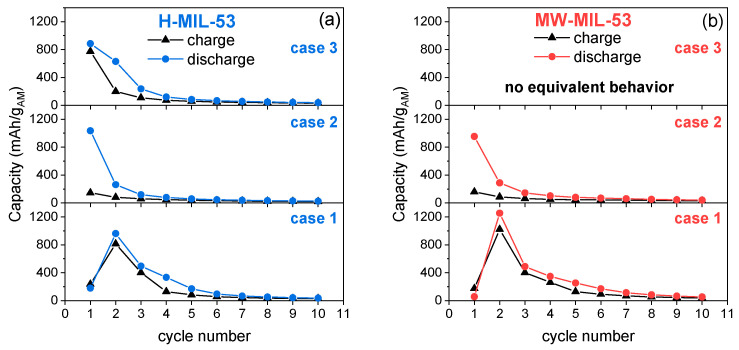
Discharge (blue or red circle) and charge (black triangle) capacities as a function of the cycle number for Li-O_2_ batteries with (**a**) H-MIL-53 or (**b**) MW-MIL-53 electrodes.

**Figure 7 materials-14-04618-f007:**
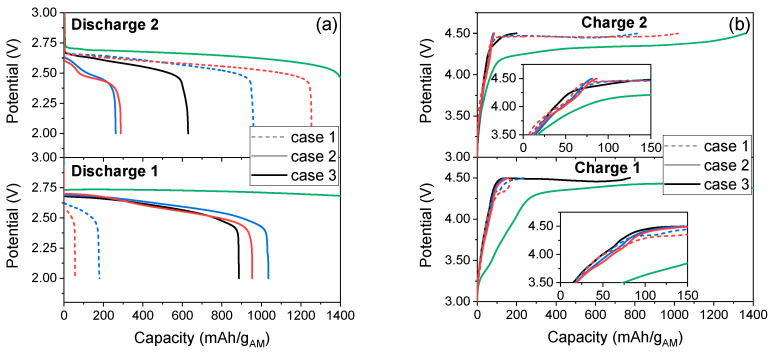
Comparison of (**a**) discharge and (**b**) charge potentials of cycle 1 and cycle 2 for Li-O_2_ batteries with H-MIL-53 (blue), MW-MIL-53(red) and Super P carbon (green) electrodes.

**Figure 8 materials-14-04618-f008:**
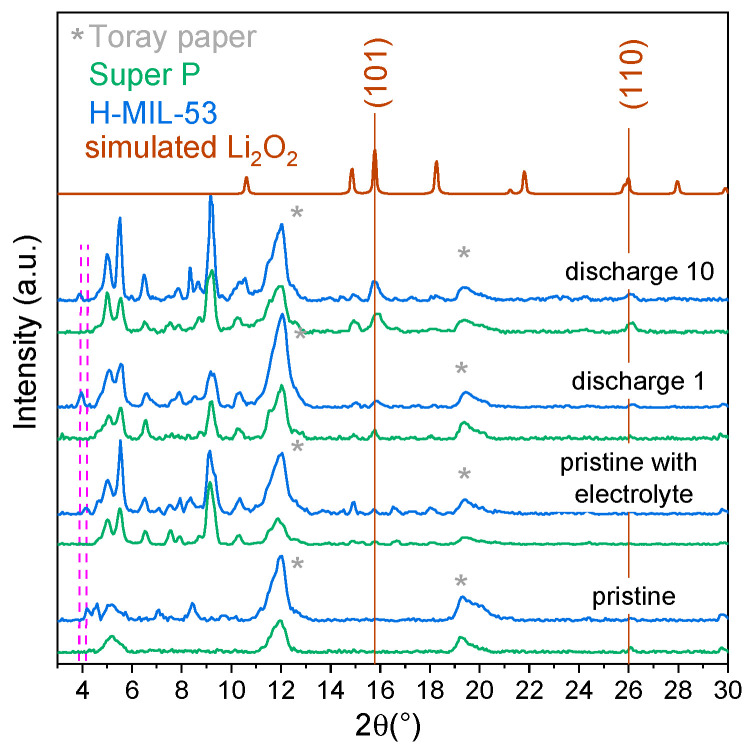
Ex situ XRD patterns (Mo-Kα) of the Super P carbon (green) and MIL-53 (blue) electrodes after 1 and 10 discharges, compared to a pristine electrode and a pristine electrode impregnated with 1M LiTFSI electrolyte. To visualize the evolution of the discharge product Li_2_O_2_, solid lines represent the Li_2_O_2_ Bragg positions (101) and (110). The peak positions attributed to MIL-53 materials are drawn in pink dashed lines.

**Figure 9 materials-14-04618-f009:**
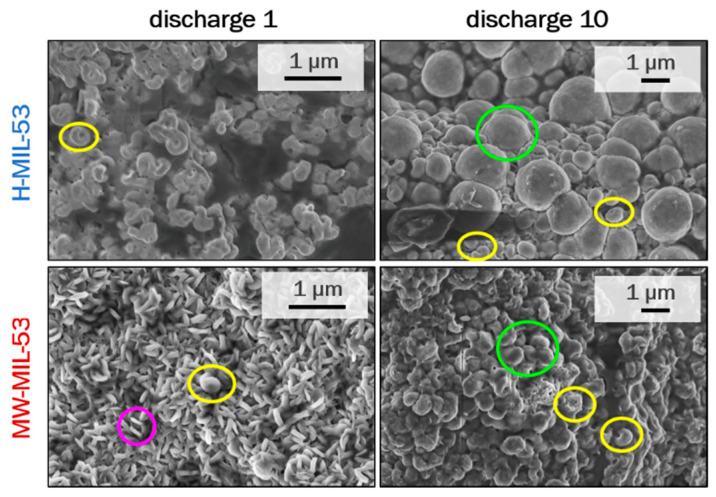
Ex situ SEM images of both MIL-53 electrodes after cycling for one and ten discharges. Stacked thin plates, small toroids and spheres of Li_2_O_2_ are circled in pink, yellow and green, respectively.

**Figure 10 materials-14-04618-f010:**
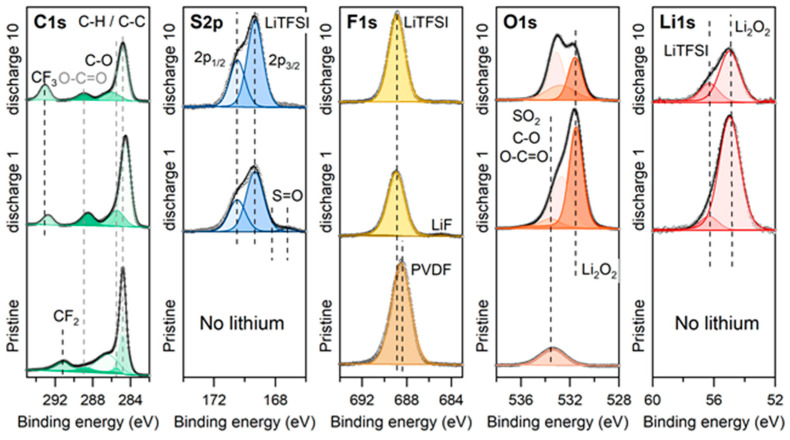
Ex situ XPS spectra of pristine and discharged MIL-53 electrodes in C 1s; S 2p, F 1s, O 1s and Li 1s regions.

**Table 1 materials-14-04618-t001:** Surface area and particle size of Super P carbon and both MIL-53 materials. The particle size of H-MIL-53 and MW-MIL-53 was evaluated using SEM images.

Material	Surface Area (m^2^/g)	External Surface Area (m^2^/g)	Particle Size
Super P carbon	52.52 ± 0.44	39.65	40 nm
H-MIL-53	1240.46 ± 2.51	52.13	2 µm
MW-MIL-53	1390.72 ± 0.43	48.31	500 nm

## Data Availability

Not applicable.

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
