# Peer review of "MIL-53 Metal–Organic Framework as a Flexible Cathode for Lithium-Oxygen Batteries"

_materials, 2021, doi:10.3390/ma14164618_

Round 1

Reviewer 1 Report

The manuscript „MIL-53 Metal-Organic Framework as a flexible cathode for lithium-oxygen batteries” by Surblé et al. is an interesting piece of work. The authors have synthesized MIL-53 in two ways, hydrothermally as well as microwave-assisted, leading to the same metal organic framework but with different particle sizes and morphology. A full investigation of the two prepared MIL-53 composites is provided in the submitted manuscript and additional information are given in the supporting information, such as powder patterns, scanning electron microscopy, thermogravimetric analysis and porosimetry. In order to test the composites as air-cathode for Li-air/Li-O2 batteries, the authors have prepared various MIL-53 composites with carbon and have investigated their electrochemical behavior, finding that the composite material is an efficient air cathode material with some limitations due to higher overpotentials for the charge and discharge processes. The manuscript is well-written, and I can recommend publication in Materials after some minor revisions.

Some minor comments:

  1. It could be helpful for the reader to include a Scheme within section 2.3, to show how the battery testing is assembled.
  2. Figure S1: While the powder pattern of H-MIL-53 and MW-MIL-53 are quite similar for the np and calcined samples, there is one additional peak in the H-MIL-53-as at ca. 17° while this is absent in the MW-form. While the ratios of all other peaks are quite similar, this can’t be an orientation effect. Can the authors provide an explanation for this?
  3. The authors nicely describe the differences in the structural transformations of MIL-53 (Figure 1), and have included the cell parameters (Table S1). According to the manuscript, the orthorhombic cell of the large pore form (lp) is Imma, while in the ESI, the ht form with the same cell volume is named Immc. This needs clarification.
  4. page 9: The authors relate the effect when the second discharge capacity is higher than the first one, and therefore the charge potential seems to be slightly lower for the hydrothermally prepared sample than for the microwave-assisted one, to arise from the particle size difference. Since it is more challenging to prepare bigger particles from the MV-method, have the authors tried to decrease the particle size of the hydrothermal compound?
  5. The ex-situ characterization is well done and clearly shows the accumulation of Li2O2 upon cycling. It would be nice to see a comparative investigation of the H-MIL-53 and the MW-MIL-53 compounds here as well.
  6. page 11: The XPS measurements reveal an additional doublet in the S2p region for the first discharge electrode. It is not entirely clear where the imide salt comes from. This needs further clarification. Since this only appears in the first cycle, can this be another reason for the limited first discharge capacity?

Author Response

We would like to thank the reviewer for their report. We modified the manuscript according to their recommendations. We hope that we have correctly replied to all their remarks. All details of our corrections are summarized in the following letter and highlighted in the revised manuscript.

We hope that the corrections will convince the reviewers to publish this work.  

Response to Reviewer 1 Comments

The manuscript “MIL-53 Metal-Organic Framework as a flexible cathode for lithium-oxygen batteries” by Surblé et al. is an interesting piece of work. The authors have synthesized MIL-53 in two ways, hydrothermally as well as microwave-assisted, leading to the same metal organic framework but with different particle sizes and morphology. A full investigation of the two prepared MIL-53 composites is provided in the submitted manuscript and additional information are given in the supporting information, such as powder patterns, scanning electron microscopy, thermogravimetric analysis and porosimetry. In order to test the composites as air-cathode for Li-air/Li-O2 batteries, the authors have prepared various MIL-53 composites with carbon and have investigated their electrochemical behavior, finding that the composite material is an efficient air cathode material with some limitations due to higher overpotentials for the charge and discharge processes. The manuscript is well-written, and I can recommend publication in Materials after some minor revisions.

Point 1: It could be helpful for the reader to include a Scheme within section 2.3, to show how the battery testing is assembled.

Response 1: A figure (Figure 1) were added in the text, showing an ECC-Air cell and a schematic representation of the Li-air battery.

Point 2: Figure S1: While the powder pattern of H-MIL-53 and MW-MIL-53 are quite similar for the np and calcined samples, there is one additional peak in the H-MIL-53-as at ca. 17° while this is absent in the MW-form. While the ratios of all other peaks are quite similar, this can’t be an orientation effect. Can the authors provide an explanation for this?

Response 2:  The pore of the as-synthesized form are filled with disordered free acid. The additional peak near 17.5° corresponds to the benzene 1,4-dicarboxylic acid. This peak is more intense for the microwave-assisted route than the hydrothermal one. The intensity of this peak depends on the acquisition and can be explained rather by more or less acid in the analysed powder than an orientation effect. We added a symbol in the graphs in order to indicate the presence of benzene 1,4-dicarboxylic acid .

Point 3: The authors nicely describe the differences in the structural transformations of MIL-53 (Figure 1), and have included the cell parameters (Table S1). According to the manuscript, the orthorhombic cell of the large pore form (lp) is Imma, while in the ESI, the ht form with the same cell volume is named Immc. This needs clarification.

Response 3: The space group for the orthorhombic cells are all Imma (n°74). The space group Immc does not exist (the others possible symmetries are Immb, Ibmm, Icmm, Imcm or Imam).

Point 4: page 9: The authors relate the effect when the second discharge capacity is higher than the first one, and therefore the charge potential seems to be slightly lower for the hydrothermally prepared sample than for the microwave-assisted one, to arise from the particle size difference. Since it is more challenging to prepare bigger particles from the MV-method, have the authors tried to decrease the particle size of the hydrothermal compound?

Response 4: No attempt have been tried to decrease the particle size using the hydrothermal method to prepare MIL-53. It is a good idea to distinguish the effect of the particle size and aspect ratio on electrochemical performances.

Point 5: The ex-situ characterization is well done and clearly shows the accumulation of Li2O2 upon cycling. It would be nice to see a comparative investigation of the H-MIL-53 and the MW-MIL-53 compounds here as well.

Response 5: Unfortunately, the RU-200B rotating anode X-ray generator was failed. We are not able to prove the presence of Li2O2 for the tenth discharge. Due to COVID crisis, it is difficult to have access to another Mo-Ka diffractometer or XPS spectrometer to confirm the nature of the discharge products for the tenth discharge. We added the XRD patterns of the first discharge in supplementary materials.

Point 6: page 11: The XPS measurements reveal an additional doublet in the S2p region for the first discharge electrode. It is not entirely clear where the imide salt comes from. This needs further clarification. Since this only appears in the first cycle, can this be another reason for the limited first discharge capacity?

Response 6:  The imide salt comes from the electrolyte LiTFSI (lithium bis[trifluoromethanesulfonyl imide]). We have replaced the sentence “most probably arising from decomposition products of the imide salt“  by “most probably arising from decomposition products of the electrolyte salt”. 

The ex situ analyses were only performed on electrodes showing a first higher discharge capacity than the second one.  Supplementary investigations could have performed to understand the reasons of the limited first discharge capacity.

Reviewer 2 Report

In this work, the authors have presented the results on the usage of MIL-53 porous structure material as an air cathode for Li-O2 batteries. The type of synthesis and parameters influence the pore size and aspect ratio. Electrochemical characteristics are tested and obtained results are promising in regard to possible application and use as cathode material.

General remark

The idea seems interesting. However, in my opinion, the manuscript consists of a lot of Figures (9 with a,b,c..d). My suggestion is to change it and make condense with better connections between different parts in the manuscript, which will result in a better outcome. Some results could be transferred additionally to the Supplementary materials.

In the light of scientific content, there are some points that I feel should be addressed and clarified:

  1. Abstract: Long and to general especially the first part. Please add more result data.
  2. Conclusions: In my opinion, it is too lengthy. Please Condense
  3. Introduction: Please add more up-to-date references, also consider maybe it is too lengthy.
  4. Please increase Figure quality. In some, the text is too small to read without zoom, fig 4 and 5
  5. Sections: Materials and methods are section 2. Please add more details about experiments performed, for example, SEM.
  6. Could you please comment on the conductivity and electrical/dielectric properties of such porous material, and does it have an effect on the cathode performances in total?
  7. I also suggest extensive editing of the English language and style. And a lot of typographical errors. Please read the text carefully and do the necessary changes.

Due to the all above mentioned, I suggest major revision.

Author Response

We would like to thank the reviewer for their report. We modified the manuscript according to their recommendations. We hope that we have correctly replied to all their remarks. All details of our corrections are summarized in the following letter and highlighted in the revised manuscript.

We hope that the corrections will convince the reviewers to publish this work.  

 Response to Reviewer 2 Comments

In this work, the authors have presented the results on the usage of MIL-53 porous structure material as an air cathode for Li-O2 batteries. The type of synthesis and parameters influence the pore size and aspect ratio. Electrochemical characteristics are tested and obtained results are promising in regard to possible application and use as cathode material.

Point 1: General remark - The idea seems interesting. However, in my opinion, the manuscript consists of a lot of Figures (9 with a,b,c..d). My suggestion is to change it and make condense with better connections between different parts in the manuscript, which will result in a better outcome. Some results could be transferred additionally to the Supplementary materials.

Response 1: We reduced the number of the figure (12 instead 15 figures in the first manscript). Moreover, the figures were simplified to better understanding. According to the reviewer 1, an additional figure was adding in order to show an ECC-Air cell and a schematic representation of the Li-air battery. The figures 4b and 4c were transferred to supplementary materials.

In the light of scientific content, there are some points that I feel should be addressed and clarified:

Point 2: Abstract: Long and to general especially the first part. Please add more result data. Conclusions: In my opinion, it is too lengthy. Please Condense

Response 2: We added more experimental results in the abstract, especially the different behaviors observed upon cyling a Li-O2 battery with both MIL-53 electrodes. We condensed the conclusion by removing general consideration mentioned at the beginning.

Point 3: Please add more up-to-date references, also consider maybe it is too lengthy.

Response 3:  We thank the reviewer for this comment. More recent reviews has been published in 2021. We reduced also the number of the reference (41 references instead 62 references). The new references were highlighted in red in the text and in the reference section.

Point 4: Please increase Figure quality. In some, the text is too small to read without zoom, fig 4 and 5

Response 4: The figures were simplified to better understanding and the police was increase.

Point 5: Materials and methods are section 2. Please add more details about experiments performed, for example, SEM.

Response 5: More details about the sample preparation before the ex situ analyses were added in the section 2.3 Battery assembly and testing and section 2.4 Characterizations after cycling.

Point 6: Could you please comment on the conductivity and electrical/dielectric properties of such porous material, and does it have an effect on the cathode performances in total?

Response 6: We added the sentence “The poor electric conductivity of MOF also increases the polarization and gives a high overpotential, leading to unsatisfactory ORR and OER reactions [33].”

Point 7: I also suggest extensive editing of the English language and style. And a lot of typographical errors. Please read the text carefully and do the necessary changes.

Response 7: We read carefully the guidelines for authors, particularly the section about Manuscript English Editing. Typographical errors such as “space” were corrected.

Round 2

Reviewer 2 Report

Dear Authors, 

thank you for making changes to the Manuscript according to suggestions.